# Predicting Key Grassland Characteristics from Hyperspectral Data

**Patrick Jackman** [1,2,*], **Thomas Lee** [3] , **Michael French** [4], **Jayadeep Sasikumar** [1,2] , **Patricia O'Byrne** [1,2],
**Damon Berry** [3] , **Adrian Lacey** [4] **and Robert Ross** [1,2]

1   School of Computing, Technological University Dublin, Grangegorman Lower, D07 XT95 Dublin, Ireland; jayadeep.kumarsasikumar@tudublin.ie (J.S.); patricia.obyrne@tudublin.ie (P.O.); robert.ross@tudublin.ie (R.R.)
2   Irish Centre for Applied AI and Machine Learning (CeADAR), Block 9/10 NexusUCD, Belfield Office Park, Clonskeagh, D04 V2N9 Dublin, Ireland
3   School of Electronic and Electrical Engineering, Technological University Dublin, Grangegorman Lower, D07 XT95 Dublin, Ireland; thomas.lee@tudublin.ie (T.L.); damon.berry@tudublin.ie (D.B.)
4   TANCO Global, Royal Oak, R21 E278 Bagenalstown, Ireland; mfrench@itanco.com (M.F.); alacey@itanco.com (A.L.)
*   Correspondence: patrick.jackman@tudublin.ie

**Abstract:** A series of experiments were conducted to measure and quantify the yield, dry matter content, sugars content, and nitrates content of grass intended for ensilement. These experiments took place in the East Midlands of Ireland during the Spring, Summer, and Autumn of 2019. A bespoke sensor rig was constructed; included in this rig was a hyperspectral radiometer that measured a broad spectrum of reflected natural light from a circular spot approximately 1.2 m in area. Grass inside a 50 cm square quadrat was manually collected from the centre of the circular spot for ground truth estimation of the grass qualities. Up to 25 spots were recorded and sampled each day. The radiometer readings for each spot were automatically recorded onto a laptop that controlled the sensor rig, and ground truth measurements were made either on-site or within 24 h in a wet chemistry laboratory. The collected data was used to build Partial Least Squares Regression (PLSR) predictive models of grass qualities from the hyperspectral dataset, and it was found that substantial relationships exist between the spectral reflectance from the grass and yield ($r^2 = 0.62$), dry matter % ($r^2 = 0.54$), sugar content ($r^2 = 0.54$) and nitrates ($r^2 = 0.50$). This shows that hyperspectral reflectance data contains substantial information about key grass qualities and can form part of a broader holistic data-driven approach to provide accurate and rapid predictions to farmers, agronomists, and agricultural contractors.

**Keywords:** ensilement; grass quality; hyperspectral reflectance; predictive models

## 1. Introduction

As the global population is now expected to rise to 9.8 billion by 2050 [1], the urgency of making the optimal use of available agricultural resources is intensifying. This is further underlined by the fact that food production resources are declining [2,3].

A prime candidate for process optimisation is the ensilement process, whereby edible and nutritious fodder is harvested in the summer months and stored for animal consumption in the winter months. This process can be optimised by knowledge of the yield and dry matter percentage of the pasture at the time of mowing [4,5], amongst other features. As a result of this and specific local issues such as the Irish fodder crises in recent years [6], the inefficiencies of the ensilement process need to be addressed to maximise yield and the quality of yield per hectare.

At the centre of the inefficiency of ensilement is a parallel labour crisis affecting the entire Irish Agri-Food industry [7]. This results in farmers having to rely upon experience

to make ensilement decisions, and thus, historical averages are applied rather than optimal choices. Technology can support an achievable route to optimal decision-making. Two recent developments have made this increasingly possible; firstly, the fall in the cost of electronics and other sensing hardware, and secondly, the expansion of machine learning and data analytics. While this paper focuses primarily on the former, there is parallel research [8] exploring the latter for this series of experiments. Thereby a robust and accurate predictive model of optimal ensilement parameters is achievable with sufficient collection of high-quality data.

In recent years, there has been a strong drive within the agricultural machinery sector to take advantage of this technological revolution and many of the global giants in this sector have spectral sensing hardware and accompanying software to provide a rapid diagnosis to the user. Prime examples include the John Deere HarvestLab [9]; the CLAAS JAGUAR [10], and the YARA N-SENSOR [11]. Thus, there has been a drive by other machinery producers to develop their own technology in this space.

The basis for using spectral reflectance as a proxy for grass qualities lies in the fact that vegetation absorbs and reflects light at various frequencies [12] and these have been used for decades to assist farm decision making [13]. More specifically photosynthesising vegetation will absorb visible and ultra-violet light [14] and approximately half of the incident infrared light [15]. Furthermore, as vegetation dies, the infrared absorption will increase and red light absorption will decrease [16], leading to a change in spectral profile and hence, dying vegetation turns yellow.

A host of studies have comprehensively examined the successful ability of spectral reflectance to act as a proxy for vegetation biomass [17–20]. A highly relevant study is that carried out by Prabhakara, Hively, and McCarty [21] that comprehensively assesses the links between a wide range of vegetation indices with biomass and ground cover in Maryland (United States) using a handheld 16 band multispectral sensor and quadrat based validation sampling. A further exemplar of the power of spectral reflectance to identify biomass is demonstrated by Agapiou, Hadjimitsis, and Alexakis [22] who were able to find subtle patterns in vegetation cover on sites of archaeological interest.

Additionally, of particular interest for biomass estimation from spectral data is the so-called 'Red Edge' region that is considered very useful in that it is more robust to soil background, saturation, and spatial heterogeneity [23,24], and thus, strong efforts should be made to gather data from this region

This leads into the seminal NDVI or Normalised Difference Vegetation Index [25], which identifies living vegetation from a small and manageable number of wavelengths. Being a dimensionless number NDVI is robust to incident natural light intensity; however, NDVI has a critical weakness in that it tends to saturate with high vegetation density [26]. Research by Schaefer et al. [17] showed that in concert with a measurement of canopy height, this saturation effect can be mitigated, and thus, some form of crop height measurement needs to be included in any sensor array where possible.

Since this seminal work, a multitude of vegetation indices have been proposed with each seeking to reveal information about a key quality from a handful of wavelengths of spectral data depending upon the desired application; a comprehensive list of these indices is provided by the Index Database [27]. The great range and variation in the wavelengths used, illustrate the need to initially carry out a hyperspectral study, so the optimal wavelengths can be identified along the lines of those carried out by Zhejiang University [28–30]. Thus, a simpler and manageable multispectral system can be constructed to capture these optimal wavelengths.

This leads to a broader question of how to best use multispectral or hyperspectral data sources to estimate other in-field qualities. This has been examined on many occasions using both proximal and long-distance sensing. While yield has already been mentioned, there is an abundance of research looking into how spectral reflectance can act as a proxy for nitrogen content, and hence, nitrates content. A study of particular relevance is that by Jarmer and Vohland [31], which used proximal spectral sensing to estimate the total

nitrogen content of summer barley. A study that took a much broader view on nutrient estimation in pasture from hyperspectral reflectance was carried out by Yule et al. [32] in New Zealand, and it demonstrated that a wide range of nutrients could be estimated by this approach. Thus, there is considerable evidence that spectral reflectance can provide substantial information on vegetation nitrate content.

Similarly, research is abundant into how spectral reflectance can act as a proxy for dry matter percentage and water stress of vegetation. Hyperspectral data were used by Mirzaie, Darvishzadeh, Shakiba, Matkan, Atzberger, and Skidmore [33] to predict vegetation moisture content. Ullah, Skidmore, Ramoelo, Groen, Naeem, and Ali [34] conducted a similar hyperspectral study and found that the water content of six common leaf types could be predicted. A further pertinent study was carried out by Casas, Riaño, Ustin, Dennison, and Salas [35], estimating foliage water content using only infrared data. Water stress detection in greenhouse vegetation by spectral reflectance was reviewed by Katsoulas, Elvanidi, Ferentinos, Kacira, Bartzanas, and Kittas [36], with a multitude of successful implementations found. Thus, the utility of spectral reflectance for estimating vegetation moisture content has been strongly demonstrated.

As stated previously, the great diversity of wavelengths chosen for the various spectral data sources illustrates the need to collect a wide variety of spectral data to allow optimal wavelengths to be mined, and thus, the optimal multispectral data sources to be identified and optimal predictive models to be derived from those subsets. The recent fall in the hardware cost of proximal hyperspectral radiometers makes their use now feasible in the scope of field-based studies. Hence, in this paper, we present a series of experiments that have been carried out to test the ability of hyperspectral datasets in the visible to near-infrared range to provide the required information.

## 2. Materials and Methods

### 2.1. Experimental Design

A twenty-two-day schedule of farm site visits was arranged through the partner company TANCO GLOBAL with farmers in the East Midlands of Ireland spread throughout late spring to early autumn 2019. The schedule sought to maximise the diversity and representativity of the grasslands based on the internal records of TANCO GLOBAL. Once on-site, on the day of the data gathering, the farmer was asked to identify regions of their grasslands that were historically high and low in moisture content and also high and low in nutrients, as well as a region that was typically average for them. Up to five samples per region were gathered with each sample 20 m from any other sample in the same region. Data collection was weather-dependent and also dependent on the performance of the rig electronics so not every region had five samples taken.

### 2.2. Sensor Rig

Engineers constructed a sensor rig shown in Figure 1 Part of this rig included an Ocean Optics Flame-S hyperspectral radiometer (Halma, Amersham, Buckinghamshire, UK) without an optical fibre attached. This device is spectrally sensitive from 350 to 1050 nm. The radiometer was set 1.5 m above ground aiming vertically downwards, and thus, measuring a disc of approximately 1.2 square metres.

Bespoke Python-based software triggered the radiometer to measure reflected light on request from the laptop controlling the sensor network. The radiometer readings were recorded with an associated timestamp and global positioning satellite (GPS) data to include their spatial and temporal context. Furthermore, the aforementioned Python software added a unique sample number to the data to link it to corresponding collected grass samples and to ensure correct archiving of the data for later processing.

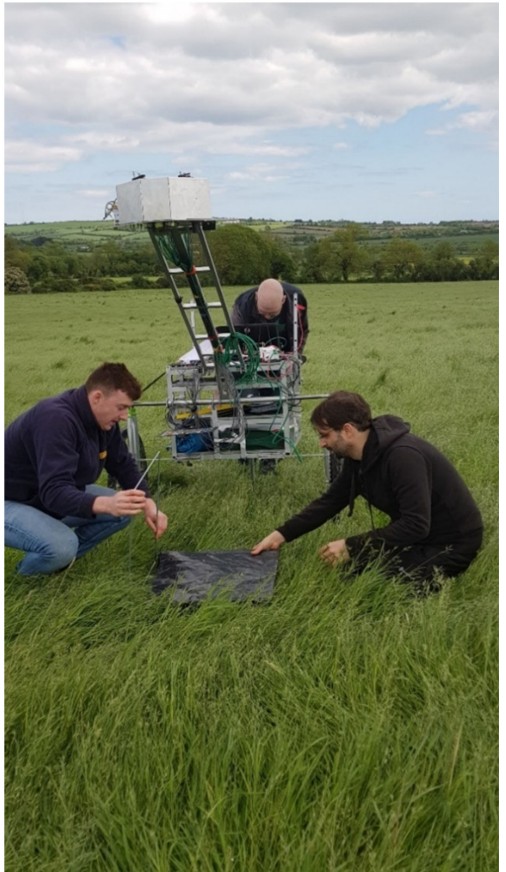 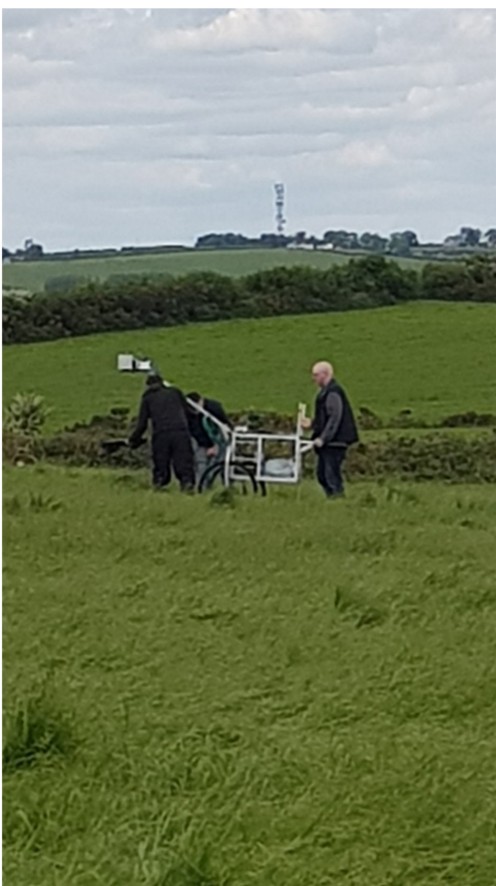

**Figure 1.** The sensor rig used for data collection.

### 2.3. Measurement of Grass Qualities

The four qualities investigated were biomass, dry matter percentage, sugars, and nitrates. While the theoretical basis for including sugars is weak, the ease of its measurement during this series of experiments led to its inclusion. Details of the measurement methodology are given below:

Biomass: A 50 cm quadrat was placed underneath the sensor rig and approximately at the centre of the viewpoint of the hyperspectral radiometer. Using a garden shear, the grass was cut to approximately 4 cm height and the cut grass was bagged into a 23 g bin liner bag. The bag was immediately transferred for on-site weighing in a Kern PCB 6000-0 scale.

Dry Matter Content: A random subsample of approximately 100 g was taken from the above bin liner and placed in a ziplock freezer bag and stored on-site in a cooler box. The samples were then brought to a food science laboratory for oven drying at 60 °C for 24 h.

Sugar and nitrates content: From the aforementioned already weighed bag of grass, four golf ball sized samples were drawn at random and crushed in a garlic crusher to release juice. In each case, four to five drops from each 'golf ball' were collected in a specimen container and closed and kept out of direct sunlight. To ensure a usable amount of juice was collected, the juice from all bags of grass from the same region was aggregated (usually 5). This corresponds to the aforementioned strategy of taking five samples from a homogeneous region of a field.

The aggregated juice for the region was then drawn into a plastic syringe before administering individual droplets onto the Brixometer and Nitrometer lenses. The first four droplets were used for the Brixometer and the next six were used for the Nitrometer. After the ten droplets had been administered the syringe was washed and dried in preparation for the next region's samples. (Not all samples had sugars and nitrates measured, as not all equipment was available at the start of the experimental program and the measurement protocols were revised after the first few weeks). For sugars, a Hanna Instruments HI96801 general-purpose refractometer was used (Hanna Instruments, Woonsocket, RI, USA). Four measurements were taken to get an average sugar value for the region. In between each measurement the refractometer lens was washed and checked to ensure that a zero reading was obtained with water only on the lens.

For nitrates, a Horiba Scientific LAQUAtwin Nitrometer (Horiba Scientific, Kyoto, Japan) was used. The instrument was prepared according to the device manual, including priming the lens at the start of the day and hard rebooting it on arrival at the farm site. Six measurements were taken to get an average nitrate value for the region. In between each measurement, the device lens was washed and checked to see that a low reading (approximately 15 to 20 ppm) was obtained with water only on the lens.

### 2.4. Model Generation

As indicated, the focus of this paper is the relationship between the hyperspectral spot average data and the key grass metrics of biomass, dry matter percentage, sugars as estimated by a Brixometer, and nitrates as estimated by a Nitrometer. For dry matter, some early readings were discarded due to a change of drying method and, for sugars, some early readings were also discarded due to a change of method. For nitrates, some early readings were discarded due to the failure to get stable readings; after assistance was obtained from Horiba Scientific, stable readings were subsequently obtained. Hence, 203 samples for biomass, 142 samples for dry matter, sugars, and nitrates were used for model creation.

Given the very large number of wavelengths (over 2000) measured and the low rank of the hyperspectral data of 11, as found by an initial principal components analysis, it was decided to decimate the data and use every 10th wavelength to create a more compact dataset and to remove redundancy. An additional motivation for decimating the dataset was that the next genus of the field spectral measurement device is to be a multispectral device using bandpass lens filters to screen incoming light. Such lens filters have a tolerance of approximately $+/- 2$ nm (ThorLabs, Newton, NJ, USA). Thus, some of the precision available in the raw hyperspectral datasets will be unusable on the next genus of the device. Notwithstanding the models were repeated without decimation for comparison purposes.

The decimated or "compressed" dataset was then mined with a slightly modified version of the genetic algorithm in Matlab (The Mathworks, Natick, MA, USA) developed by Jackman, Sun, Allen, Valous, Mendoza, and Ward [37] that sought the wavelengths leading to the best multilinear regression with a condition number of less than 1000. This method seeks to cope with the low rank of the spectral datasets by identifying the wavelengths that carry the strongest correlation with the response variables. As there is no a priori reason to suspect one wavelength over another a genetic algorithm provides a computationally feasible platform for freely sifting through the potential combinations of wavelengths that might best correlate with the response variable. The genetic algorithm optimisation criterion was to minimise the negative of the subset multilinear regression correlation coefficient.

One drawback of genetic algorithms is that they can overfit and converge on a subset that is relying on noise and fluctuations to account for model variance; as a safeguard against overfitting, the condition number of a regression matrix estimates the noise vulnerability of the regression model. Thus, any prospective subset of wavelengths leading to a regression matrix condition number greater than 1000 was rejected as unfit by the genetic algorithm (condition number of 1000 approximately corresponds to 3 lost digits).

Using the mined variables, PLSR models were built with the multivariate software Unscrambler (Camo Analytics, Oslo, Norway). The mined variables are shown in Table 1 By coincidence, twelve wavelengths were found for each of the four qualities which closely matches the rank of the dataset suggested by the principal components analysis.

**Table 1.** Wavelengths in nanometres are identified as important for each target variable.

| Feature | Wavelengths in Nanometres | | | | | | | | | | | |
|---------|---|---|---|---|---|---|---|---|---|---|---|---|
| Biomass | 450 | 490 | 519 | 544 | 587 | 591 | 647 | 653 | 664 | 698 | 806 | 1020 |
| Dry% | 398 | 413 | 431 | 555 | 608 | 629 | 660 | 698 | 860 | 881 | 902 | 962 |
| Sugars | 394 | 512 | 541 | 601 | 619 | 657 | 677 | 761 | 816 | 819 | 916 | 1015 |
| Nitrates | 394 | 548 | 615 | 629 | 677 | 691 | 728 | 731 | 761 | 809 | 819 | 993 |

## 3. Results

The coefficient of determination and the root mean square error of prediction divided by the target standard deviation were calculated based on the average of five random 10-fold splits. Whereby, the data was on five occasions split randomly into 10 segments for 10-fold cross-validation. The average of the five PLSR models was chosen as the result. Results are shown in Table 2 as coefficients of determination and root mean square error of prediction divided by the target standard deviation.

**Table 2.** Model metrics for each target variable.

| 10-Fold Cross Validation | Biomass | Dry% | Sugars | Nitrates |
|--------------------------|---------|------|--------|----------|
| Model1 Coefficient of Determination | 0.624 | 0.514 | 0.550 | 0.489 |
| Model2 Coefficient of Determination | 0.607 | 0.522 | 0.548 | 0.494 |
| Model3 Coefficient of Determination | 0.628 | 0.548 | 0.498 | 0.498 |
| Model4 Coefficient of Determination | 0.614 | 0.508 | 0.476 | 0.476 |
| Model5 Coefficient of Determination | 0.622 | 0.538 | 0.471 | 0.471 |
| Average Coefficient of Determination | 0.62 | 0.53 | 0.54 | 0.49 |
| Average RMSEP/$\sigma$ | 0.61 | 0.68 | 0.67 | 0.72 |

The results show that the hyperspectral radiometer contains substantial and important data sources related to key grassland metrics. This information can inform the further ensilement decision-making processes [38]. However, by themselves, these results do not offer highly accurate predictions.

The repetition of the modelling with the raw hyperspectral rather than the decimated datasets did not improve the models and actually disimproved the models ($r^2$ for biomass = 0.48, $r^2$ for dry matter % = 0.32, $r^2$ for sugars = 0.27, and $r^2$ for Nitrates = 0.2). Additionally, the software used to run the genetic algorithm search repeatedly warned of rank deficient regression matrices in the prospective candidate solutions. This suggests that with such a low rank of the raw dataset, the redundancy in the raw data was strongly impeding the genetic algorithms search for the optimal subset.

The results further show that a small subset of wavelengths of approximately 10–15 does carry most of the predictive information and that a transition to a faster and less burdensome multispectral approach is likely and feasible. Conversely, it is not feasible to build a traditional vegetation index to express this information due to the complex predictor inter-relationships. This will vindicate the parallel research that uses complementary multispectral channels to generate hyperlocal images of the same grassland [8]. Additionally, the wavelengths that survived the genetic selection indicate that a broad spectrum of channels is needed to describe the model variation from blue 400 nm to 1000 nm infrared.

## 4. Discussion

The results for grassland biomass estimation compare well with the work of Xiaoping et al. [39], who used remote hyperspectral sensing to estimate total biomass on Chinese Rangelands, where coefficients of determination up to 0.74 were found to be possible with a multitude of hyperspectral indices. Similar results were obtained by Gao et al. [40], with a coefficient of determination of 0.70 for total biomass on Tibetan plateaus.

The results for nitrates are similar to the PLSR model of Pullanagari et al. [41], which predicted total nitrogen content with a coefficient of determination of 0.54 in New Zealand. However, the benefits of using deep neural networks rather than multivariate statistics were strongly underscored by a convolutional neural network that achieved a coefficient of determination of 0.72 on the same dataset. This vindicates the parallel research aforementioned [8] that is being carried out on the counterpart images from this current series of experiments and is focusing heavily on the use of deep learning neural networks to predict grassland traits.

The results for dry matter percentage or by the default water content of the grass pasture do not compare very well to studies that could avail of Short Wave Infrared (SWIR) hyperspectral data such as that by Mirzaie et al. [33]. This study was able to find a very accurate PLSR model accounting for up to 94% of the model variance and this would strongly indicate that any spectral device aiming to provide very accurate measurements of dry matter percentage needs to be able to acquire SWIR data. Unfortunately, SWIR sensors and especially SWIR cameras are very expensive, which can make them uneconomic choices for a wide range of applications; however, there is a long-term trend of falling costs in electronic hardware and perhaps in coming years prices may fall sufficiently to address this.

As already mentioned above, parallel research on image processing is currently underway and the outcomes of that analysis can be synergised with these results to see if a holistic modelling approach can create more accurate and robust models and algorithms. Additional research funding is being sought to gather further datasets, and if funding is awarded, this will take place. Laboratory measurement of chlorophyll would be included in any such future study as this important feature has been proven to be related to biomass spectral reflectance [42]. Similarly, laboratory measurements of crude protein and digestible and non-digestible matter can be easily included as suggested by Thulin et al., [43]. Furthermore, future research will use dynamic imaging for data collection with the sensor box mounted onto a working mower.

Such further research to encompass multiple growing seasons would be of great value to ensure that the models and data gathering devices developed at the end of the next genus of the technology will be highly robust and prepared for the seasonal and year-to-year variations. Similarly, the strong global market presence of the partner company TANCO GLOBAL offers opportunities for additional data gathering worldwide to maximise the regional diversity and representativity of the available datasets when the next genus of the device is completed.

Another important question for further refinement of the methods developed is using image processing to discriminate between useful biomass and weeds and the aforementioned parallel research [8] is already utilising image processing algorithms to characterise grasslands. While the current study used grasslands with a very low weed density, future studies may encounter much higher weed densities and this confounding factor will need to be dealt with. There has been a multitude of recent studies on segmenting weeds from useful biomass using image processing indicating this is a challenge that can be easily met [44–47].

## 5. Conclusions

The hyperspectral radiometer has proven to be a valuable and important data source concerning key grassland qualities and, on its own, can lead to models that show a substantial correlation with independently measured metrics. However, it has fallen short

of being a definitive source and requires complementary data sources to reach that level. Furthermore, the acquisition of additional datasets over multiple growing seasons and multiple diverse and representative regions will lead to more robust and more useful predictive models. The data analysis has also shown that a small subset of well-chosen wavelengths can condense the predictive power of the radiometer into 10–15 wavelengths that opens up the possibility of a much faster and dynamic multispectral data acquisition technology.

**Author Contributions:** Conceptualization, P.J., A.L. and R.R.; Data curation, T.L. and P.O.; Formal analysis, P.J. and P.O.; Funding acquisition, P.J., A.L. and R.R.; Investigation, P.J., T.L., P.O., D.B., and R.R.; Methodology, P.J., M.F., D.B. and R.R.; Project administration, P.J.; Software, P.J., T.L. and J.S.; Supervision, D.B., A.L. and R.R.; Writing—original draft, P.J.; Writing—review & editing, P.J. All authors have read and agreed to the published version of the manuscript.

**Funding:** This research was carried out with funding from the Enterprise Ireland Innovation Partnership Scheme (IP2018/0728-GREENEYES) and TANCO GLOBAL.

**Institutional Review Board Statement:** Not Applicable.

**Informed Consent Statement:** Not Applicable.

**Data Availability Statement:** Data available on request due to restrictions.

**Conflicts of Interest:** The authors declare no conflict of interest.

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
