# Peer review of "Predicting Key Grassland Characteristics from Hyperspectral Data"

_agriengineering, doi:10.3390/agriengineering3020021_

Round 1
Reviewer 1 Report
I think the main problem with the paper is that the presentation of the results is somewhat on the run. In this way the conclusions can be questioned. Authors should give more space to the description and presentation of results. Another issue that needs to be clarified is the extent to which the conclusions may be relevant if based on one-year measurements. Finally, decimating the data can again raise questions about the relevance of the findings, because today's studies do not rely on little data, but rather rely as much as possible on a large amount of data.
Author Response
"Authors should give more space to the description and presentation of results"
A revised results section is included in the new version of the paper
"Another issue that needs to be clarified is the extent to which the conclusions may be relevant if based on one-year measurements."
This issue is integrated into the conclusions and the long term need for multi-year and multi-site datasets is recognised.
"Finally, decimating the data can again raise questions about the relevance of the findings, because today's studies do not rely on little data, but rather rely as much as possible on a large amount of data."
Further reasoning and justifications for decimating the raw data is provided in the new version of the paper
Reviewer 2 Report
Dear Authors,
Please receive the following comments on your manuscript "Predicting Key Grassland Characteristics from Hyperspectral Data".
The research is original and presents valuable field measurements, including very sophisticated hyperspectral measurements and a sound establishment of a ground-truth, fit for modelling.
In reading your manuscript, several methodological issues were detected, which you may want to consider to improve your publication. Kindly consider the following comments and suggestions.
* Section 2.1 can profit from a more thorough description of the representativeness of the measuring sites, the methods for selection them and the recommendations given by the farmers. A note on the differences in Spring, Summer and Autumm would be also helpful.
* Line 171: Please explain how the four replicates were taken. Were they sampled from the five-aggregated samples in the syringe?
* Line 178: Please explain about the replicates (see previous point).
* Section 2.4 would gain from a more detailed explanation of how and why certain wavelengths selected as predictors using Multilinear Regression (and the meaning of the 1000 condition number), to be later used in a PLSR model.
* Also regarding model construction, if there was need to:
- Decimate the data, instead of using the full potential of a very good measuring instrument
- Fix the number of predictors to 12 for all target variables (this decision seems somewhat arbitrary)
* L 294: A description of the algorithm, constrains and optimization objectives would be very helpful. Also if it was implemented by the authors themselves and what was the target of the genetic optimization. Was the number of 12 wavelengths hard-coded in the genetic algorithm?
* Format: Please make sure that the tables are not broken between pages (L.200)
* Line 207: The estimation of the model error seems appropriate and good. I suggest to create and callibrate a single PLSR model (instead of taking the average - is it the average of each parameter in the model?) with all data after an error estimate was calculated. This is valid when cross-validation was used. Otherwise, it would be good to show the results of the five trained models. It is not clear that the average (of the single parameters?) would perform as well as the individual models. This might improve the results in Table 2.
* L. 221: "The results further show that a small subset of wavelengths of approximately 10-15 doescarry most of the predictive information" → I disagree that this claim can be done, as no comparison with more complex models was carried out.
* Section 4 needs to be expanded to compare the results of the current research with those in the literature. In its current form, lines 231 to 249 could be included either in Section 5 or in an additional section "outlook" or "further work".
* Section 1 gives an excellent overview of the field. If Section 4 can mirror its quality and depth, the article will be of much more interest to the community
Kind regards
Author Response
*"Section 2.1 can profit from a more thorough description of the representativeness of the measuring sites, the methods for selection them and the recommendations given by the farmers. A note on the differences in Spring, Summer and Autumm would be also helpful."
Further details are provided on the representativeness of the sites; however given that season to season variation was not integrated into the experimental design I feel it would be hazardous to draw conclusions from any season to season variation observed as this could be due to confounding factors. Although in a longer term study it may be possible to design the experiments for seasonal variation.
"Line 171: Please explain how the four replicates were taken. Were they sampled from the five-aggregated samples in the syringe?"
Additional information clarifying this point is provided in the new version.
"Line 178: Please explain about the replicates (see previous point)."
Additional information clarifying this point is provided in the new version.
"Section 2.4 would gain from a more detailed explanation of how and why certain wavelengths selected as predictors using Multilinear Regression (and the meaning of the 1000 condition number), to be later used in a PLSR model."
Additional information is provided in the new version.
"Also regarding model construction, if there was need to:
- Decimate the data, instead of using the full potential of a very good measuring instrument
- Fix the number of predictors to 12 for all target variables (this decision seems somewhat arbitrary)"
Reasoning and justifications for decimating the dataset is provided in the new version. Clarification that the genetic algorithm was free to choose as many wavelengths as it wished is provided. The outcome of 12 wavelengths was co-incidental and closely matches the rank of 11 found by a Principle Components Analysis of the raw hyperspectral dataset.
"L 294: A description of the algorithm, constrains and optimization objectives would be very helpful. Also if it was implemented by the authors themselves and what was the target of the genetic optimization. Was the number of 12 wavelengths hard-coded in the genetic algorithm?"
Further information is provided on this point in the new version of the paper.
"Format: Please make sure that the tables are not broken between pages (L.200)"
This is fixed in the new paper.
"Line 207: The estimation of the model error seems appropriate and good. I suggest to create and callibrate a single PLSR model (instead of taking the average - is it the average of each parameter in the model?) with all data after an error estimate was calculated. This is valid when cross-validation was used. Otherwise, it would be good to show the results of the five trained models. It is not clear that the average (of the single parameters?) would perform as well as the individual models. This might improve the results in Table 2."
Results for the 5 individual PLSR models is provided in the new version showing them all very close to the average figure.
"L. 221: "The results further show that a small subset of wavelengths of approximately 10-15 doescarry most of the predictive information" → I disagree that this claim can be done, as no comparison with more complex models was carried out."
The models were repeated with all of the wavelengths and there was no improvement over the decimated data models. In fact the genetic algorithm struggled to converge due to the low rank of the dataset. The out come of the PCA test suggesting a rank of 11 is strong evidence supporting this statement.
"Section 1 gives an excellent overview of the field. If Section 4 can mirror its quality and depth, the article will be of much more interest to the community"
A revised section 4 is provided in the new version.
"Section 4 needs to be expanded to compare the results of the current research with those in the literature. In its current form, lines 231 to 249 could be included either in Section 5 or in an additional section "outlook" or "further work".
Comparisons with recent results from similar studies is provided in the new version.
Round 2
Reviewer 1 Report
Although the authors do not respond to all my comments, I believe that the paper can be published as it is.